# Discovery and Transcriptional Profiling of *Penicillium digitatum* Genes That Could Promote Fungal Virulence during Citrus Fruit Infection

**DOI:** 10.3390/jof10040235

**Published:** 2024-03-22

**Authors:** Paloma Sánchez-Torres, Luis González-Candelas, Ana Rosa Ballester

**Affiliations:** 1Instituto Valenciano de Investigaciones Agrarias (IVIA), Centro de Protección Vegetal y Biotecnología, Moncada, 46113 Valencia, Spain; 2Food Biotechnology Department, Instituto de Agroquímica y Tecnología de Alimentos (IATA), Consejo Superior de Investigaciones Científicas (CSIC), Catedrático Agustín Escardino Benlloch 7, Paterna, 46980 Valencia, Spain

**Keywords:** citrus, pathogenicity, *Penicillium digitatum*, postharvest, virulence, SSH, CWDE, regulatory proteins

## Abstract

Green mold caused by *Penicillium digitatum* (Pers.:Fr.) Sacc is the most prevalent postharvest rot concerning citrus fruits. Using the subtractive suppression hybridization (SSH) technique, different *P. digitatum* genes have been identified that could be involved in virulence during citrus infection in the early stages, a crucial moment that determines whether the infection progresses or not. To this end, a comparison of two *P. digitatum* strains with high and low virulence has been carried out. We conducted a study on the gene expression profile of the most relevant genes. The results indicate the importance of transcription and regulation processes as well as enzymes involved in the degradation of the plant cell wall. The most represented expressed sequence tag (EST) was identified as PDIP_11000, associated with the FluG domain, which is putatively involved in the activation of conidiation. It is also worth noting that PDIP_02280 encodes a pectin methyl esterase, a cell wall remodeling protein with a high expression level in the most virulent fungal strains, which is notably induced during citrus infection. Furthermore, within the group with the greatest representation and showing significant induction in the early stages of infection, regulatory proteins (PDIP_68700, PDIP_76160) and a chaperone (PDIP_38040) stand out. To a lesser extent, but not less relevant, it is worth distinguishing different regulatory proteins and transcription factors, such as PDIP_00580, PDIP_49640 and PDIP_78930.

## 1. Introduction

Citrus production is among the most highly valued in the fruit industry, evidencing the importance of this fruit in international agricultural trade [1]. Citrus fruits are prone to infection by various pathogens during postharvest storage and transportation, leading to substantial economic losses [2,3]. Postharvest diseases in citrus fruits caused by *Penicillium* are one of the most serious, with losses that, in some serious cases, can reach 90% [2]. Among them, the losses caused by *Penicillium digitatum* (Pers., Fr.) Sacc., known as green mold, is the most devastating. This necrotic fungus enters fruit tissues through wounds and has a short life cycle of 3 to 5 days but produces a large number of spores that could initiate a new life cycle [4], spreading the infection exponentially. Therefore, effective control measures are crucial for preventing significant losses, particularly during the initial stages of infection.

Currently, the control methods used during the postharvest of citrus fruits are not as effective due to the emergence of resistance to fungicides [5]. While new studies have been conducted to improve control methods [6,7], it is necessary to design and tackle a strategy change that allows for new targets to be addressed. In that sense, factors that could affect pathogen pathogenicity/virulence have acquired special relevance.

Phytopathogenic fungi employ a diverse variety of infection tactics to evade plant defence mechanisms, penetrate and invade host tissues, to finally provoke damage. The crucial factor that determines whether the host resists or the pathogen invades is the timing of spore germination at the site of penetration, which is achieved through the perception of various host-derived signals [8]. Therefore, virulence can be considered the grade of harm inflicted on the host. Virulence factors can be considered pathogen constituents that are not essential for growth but promote disease [9]. Virulence in plant pathogenic fungi is regulated by a network of cellular pathways that respond to signals during host–pathogen interactions [10,11]. 

Virulence in fungal–plant pathogens can be achieved through numerous mechanisms [2,12,13,14]. In the case of the genus *Penicillium*, the initiation of infection occurs in wounds on the surface of the fruit and is influenced by nutrients and volatiles that trigger spore germination [15]. Understanding the mechanisms underlying virulence in postharvest pathogenic fungi, especially in *Penicillium*-fruit interactions, has recently increased. However, much remains unknown about the mechanisms driving pathogen virulence and the factors involved.

In terms of *P. digitatum*–citrus interaction, most research has focused on studies addressing the functional analysis of specific genes [2,16,17,18,19,20,21,22,23,24,25,26]. Previous studies related to virulence in *P. digitatum* from a broader perspective have been carried out at stages where the infection was already patent and have mostly implicated genes coding for fungal proteases and plant cell wall-degrading enzymes [18,27,28]. However, no analysis of *P. digitatum*’s virulence mechanism has been conducted so far at such early stages (24 h) to identify the genes that trigger the infection of this pathogen. In this work, we employed subtractive suppression hybridization (SSH) to investigate the potential mechanisms contributing to the virulence of *P. digitatum*. Our focus was on identifying key genes active in the initial stages of the interaction with citrus fruits, aiming to determine their role in influencing the progression or inhibition of infection.

## 2. Materials and Methods

### 2.1. Microorganisms 

Two *Penicillium digitatum* isolates were used in this work: strain Pd1 (CECT20795), a highly virulent and fungicide-resistant strain, and Pd149 (CECT2954), a low virulent and fungicide-sensitive strain provided by the Spanish Type Culture Collection (CECT) [29,30]. 

Conidia were obtained from 1-week-old PDA plates by scraping them with a sterile spatula and transferring them to sterile water. Conidia were filtered, titrated with a hemacytometer, and then adjusted to the desired final concentration.

For propagation material and plasmid storage, *Escherichia coli* DH5α was used. Cultures were grown in LB agar plates or LB broth amended with 100 μg/mL of ampicillin at 37 °C.

### 2.2. Molecular Manipulations

The genomic DNA of *P. digitatum* strains was extracted as previously described by Marcet-Houben et al. [31]. All PCR templates obtained were purified using the Ultra Clean TM PCR Clean-up (MoBio, Solana Beach, CA, USA).

The total RNA from the spores or mycelium of *P. digitatum* was obtained from frozen tissue using Trizol (Invitrogen) following the recommendations of the manufacturer. The total RNA during fruit infection was extracted from fruit peel discs, as described previously by López-Pérez et al. [18]. Poly(A)^+^ RNA was separated from the total RNA using the Dynabeads^®^ mRNA Purification™ kit according to the manufacturer’s protocol (Invitrogen, Carlsbad, CA, USA).

### 2.3. Construction of a Subtracted cDNA Library

cDNA synthesis and the SSH procedure [32] were conducted using the PCR-Select™ cDNA Subtraction kit (Clontech, Palo Alto, CA, USA) according to the protocol supplied by the manufacturer. Both *P. digitatum* strains (10^5^ c/mL) were grown in 24-well culture plates in the presence of 5 mm diameter orange discs containing albedo and flavedo (5 discs per well) at 25 °C for 24 h with 2 mL of sterile water. The controls were performed by growing fungal strains without orange discs using 2 mL PDB and the same growth conditions. We used RNA from *P. digitatum Pd149*-infecting orange discs tissue at 1 day post inoculation (dpi) as a ‘driver’ and RNA from *P. digitatum* Pd1-infecting orange discs tissue at 1 dpi as a ‘tester’. One microgram of poly(A)^+^ RNA from the ‘tester’ and ‘driver’ was used for cDNA synthesis. The subtracted cDNA fragments were cloned into the pCRII (Invitrogen) vector and then transformed into *E. coli* DH5α competent cells via electroporation.

### 2.4. cDNA Sequencing and EST Analysis

The subtracted cDNA library, denoted as VPdS, was sequenced using 454-FLX Titanium technology with a GS Junior from Roche to perform a global transcriptomic analysis of putative fungal virulence genes. The obtained sequences were trimmed off the vector and adaptor sequences with the program Newbler 2.6. Processing and mapping of the ESTs to the *P. digitatum* isolate Pd1 [31] genome were performed with the CLC Genomics Workbench v 8.0.2 (Qiagen, Valencia, CA, USA). The UniProt database was used to functionally annotate unknown proteins [33]. GO annotation of unigenes was conducted with the program Blast2GO [34]. Functional enrichment (GO, KEGG Kyoto Encyclopedia of Genes and Genomes-and KEGG-BRITE) analyses were performed with WebMGA [35].

Similarity searches against DNA/protein sequence databases were conducted employing the BLAST programs [36].

### 2.5. Fungal Infection

Infection experiments were performed using freshly harvested oranges (*Citrus sinensis*) that were injured at four places around the equatorial axis and infected with 10 μL of a conidia suspension adjusted to 10^5^ conidia/mL. They were kept at 20 °C and 90% RH. The infection assays were carried out twice with three replicates of five fruits each. Mock-inoculated fruits were used as a control. The samples for relative gene expression were taken at 1, 2, and 3 days post inoculation (dpi). The samples for virulence analysis were evaluated at 3, 4, 5, 6, and 7 dpi. The evolution of the infection was determined using two parameters: the intensity of the infection measured by the percentage of infected fruits and the severity of the disease, using the diameter of the macerated tissue.

### 2.6. Quantification of Relative Gene Expression Using RT-qPCR

Fungal strains were grown in potato dextrose broth (PDB) or potato dextrose agar (PDA). The cultures were incubated at 25 °C for 1, 2, or 3 days (liquid cultures) depending on the subsequent use or up to 1 week (solid media). Samples used for gene expression were obtained using the Trizol method (Ambion Inc., Austin, TX, USA) for RNA extraction from *P. digitatum* frozen mycelium. The extraction of the total RNA from the infected samples was processed as reported previously [18]. Genomic DNA from the total RNA was removed using Turbo RNA-free DNAse (Ambion Inc., Austin, TX, USA). The complete elimination of DNA was confirmed by the absence of amplification of a 500 bp fragment of the β-tubulin coding gene of *P. digitatum* with oligonucleotides PS24-PS25 [29]. 

PrimeScript™ RT reagent Kit (Takara Bio Inc. CA, USA) was used for the synthesis of the first strand of cDNA in a 20 μL reaction, following the instructions of the manufacturer. Quantitative PCR was carried out as stated before [22]. 

The experimental values obtained were an average of two repetitions of three biological replicates. The oligos used for each EST and those for the gene coding for the fungal β-tubulin (qTubF-qTubR) (ADF32079.1), which was used as a reference gene, are indicated in Appendix A. LightCycler 480 Software, version 1.5 (Roche Diagnostics), was used for cycle quantification. The primer melting temperature allowed for the selection of each primer set for a specific amplification. The Relative Gene Expression (‘RGE’) procedure was carried out as stated before [30].

Expression analyses were graphically represented with the SRplot package (http://www.bioinformatics.com.cn/srplot, accessed on 15 December 2023).

### 2.7. Statistical Analysis 

Significant differences were evaluated using analysis of variance (ANOVA) with SAS software 9.4M8 (SAS Institute Inc., Cary, NC, USA). Statistical significance was defined as *p* < 0.05; when the analysis was statistically significant, Tukey’s test for separation of means was performed. 

## 3. Results

### 3.1. Evaluation of Fungal Strains

Pd1 and Pd149 strains were grown on PDA plates, and differences in sporulation were observed since Pd149 reached full sporulation after 10 days while Pd1 after 7 days (Figure 1). The infection of different fungal strains Pd1 and Pd149 used in this study was carried out in ‘Navelina’ orange fruits to determine the differences displayed in terms of fungal virulence. The analysis of the infectivity showed that the disease incidence of Pd149 was almost 50% lower than that of Pd1, and disease severity was also lower at all assessed times (Figure 1).

### 3.2. Sequence Analysis of the VPdS Library

The substrated VPdS cDNA library contained 81,354 reads, from which 79,467 were aligned. After sequence assembling using Newbler 2.6., the VPdS cDNA library contained 792 singletons, 123 contigs, 87 isogroups, and 188 isotigs (Table 1). After manual curation and mapping against the *P. digitatum* genome sequence, the contigs were assigned to 78 *P. digitatum* genes and 20 contigs were not mapped against any annotated gene. Moreover, 12 contigs did not map against any region in the genome.

The most represented EST (PDIP_11000) with 12,951 total reads corresponds to a *P. digitatum* Pd1 hypothetical protein. Analysis with UniProt showed similarity to the FluG domain protein. Interestingly, among the 20 most abundant unigenes, there were five genes coding hypothetical proteins with unknown function. Additionally, genes coding for a CYP51-like protein (PDIP_01820), a gene (PDIP_40660) encoding a glycerol uptake facilitator, *P. digitatum* genes presumably related to pathogenesis, such as a pectin esterease (PDIP_02280), and a C6 transcription factor (PDIP_33600), were identified. The remaining 11 unigenes included a *P. digitatum* ribosomal protein. Regulation pathways were also represented by two C6 transcription factors (PDIP_33600 and PDIP_00580): a protein serine/threonine kinase (Ran1) (PDIP_28970), a regulator of G protein signaling domain (PDIP_68700), a cAMP-independent regulatory protein Gti1/Pac2 (PDIP_76160), a transcription factor (Snd1/p100) (PDIP_49640), and a calcium/calmodulin-dependent protein kinase (PDIP_78930). Another group included different types of transporters: a glycerol transporter (PDIP_40660) and three metal transporters such as CorA family metal ion transporter (PDIP_22490), iron copper transporter (PDIP_41920) and iron transport multicopper oxidase Fet3 (PDIP_59000). Several unigenes were involved in enzymatic processes, such as dehydrogenases (PDIP_08960, PDIP_49790), acyltransferase (PDIP_80820), oxidase (PDIP_47340), permease (PDIP_83140), glucanase (PDIP_21000), lyase (PDIP_06990), reductases (PDIP_07120, PDIP_49590), and transaminase (PDIP_65080) (Appendix A).

### 3.3. Functional Annotation

The transcriptional responses identified in the VPdS library were further analyzed through functional enrichment analysis of GO terms (Figure 2). The sequences were classified into the major functional categories of biological processes. Those sequences were enriched in biological processes related to translation, amide biosynthesis, and peptide biosynthesis, among others.

EuKaryotic Orthologous Groups (KOG) class enrichment analysis showed the top 18 terms that were significantly enriched, highlighting translational modification, post-translational modification protein turnover, and signal transduction mechanisms (Figure 3A). In the KEGG pathway analysis, ribosomes, metabolic pathways, and biosynthesis of secondary metabolites were the most represented KEGG pathway (Figure 3B). Among the BRITE categories, enzymes, ribosomes, transporters, and exosomes were the most prominent (Figure 3C).

### 3.4. Evaluation of Gene Expression Profiling in Orange Discs

Among all the genes identified in the VPdS cDNA library, a total of 21 genes were selected considering either their representation, determined by the number of reads, or their possible function based on their putative function determined by its homology in databases. In this sense, the first twelve genes were selected based on their representation in a number of reads, and the remaining nine genes had values from medium to low in terms of the number of reads but with apparently interesting functions (Table 2).

The hypothetical function was assigned based on the genome annotation of *P. digitatum* Pd1 [29], and in those cases where it was only identified as a “hypothetical protein”, as in PDIP_11000, PDIP_49600, PDIP_76190 and PDIP_64910, a more exhaustive analysis was carried out using the recent genome of *P. digitatum* PDW03 [37] and NCBI database.

To determine the efficiency of the VPdS library in terms of selecting genes induced early during the interaction of the virulent isolate Pd1 with orange discs as compared to the less virulent P149, we conducted a gene expression study. Additionally, we examined the control samples grown in vitro in 24-well plates with PDB medium for 24 h. The results presented in the heat map showed that all the genes evaluated had a higher gene expression in Pd1 compared to Pd149 during their interaction with orange discs, thus demonstrating the efficiency of the subtraction. In four cases (PDIP_11000, PDIP_28790, PDIP_33600, and PDIP_49600), the expression level in Pd1 was higher in PDB than during disc interaction and in PDIP_68700 rate of expression was the same for Pd1. Thus, expression analysis showed that most genes in the low-virulent Pd149 isolate also had higher expression in the orange disc interaction than during growth on the PDB medium. Nevertheless, five genes (PDIP_22490, PDIP_64910, PDIP_66160, PDIP_68700, and PDIP_74620) showed a lower expression rate in the Pd149 isolate during disc interaction compared to growth in PDB. (Figure 4).

It is also worth noting that the expression magnitude was different according to each gene, with PDIP_05570 being the most abundant, followed by PDIP_11000, PDIP_76190, PDIP_02280, PDIP_06990, and PDIP_78930.

### 3.5. Analysis of Gene Expression 

In accordance with what was previously observed, a more exhaustive study of the gene transcription profile was carried out for the 21 selected genes, analyzing their expression over time (1 to 3 dpi) both in in vitro cultures and during infection of orange fruits by the two *P. digitatum* strains (Figure 5).

Heat map and ballon plot showed elevated transcriptional rate in all genes of the Pd1 strain analyzed compared to Pd149 during orange infection at 1 dpi, indicating higher expression levels in the early stages and corroborating the efficiency of the VPdS cDNA subtraction library (Figure 5).

The heat map analysis showed that 10 of the selected genes (PDIP_02280, PDIP_68700, PDIP_38040, PDIP_76190, PDIP_00580, PDIP_64910, PDIP_01590, PDIP_49640, PDIP_78930, and PDIP_06990) displayed the highest expression in the virulent isolate Pd1 during infection of oranges at 1 dpi, while PDIP_33600 peaked at 2 dpi. In two cases (PDIP_28970 and PDIP_66160), the highest induction took place during the in vitro growth of Pd1, and only in the case of PDIP_76160 was the induction surpassed by Pd149 during in vitro growth (Figure 5A). However, there were variations in both the magnitude of the expression level and the overall trend of gene expression for each specific gene, as shown in the ballon plot (Figure 5B).

For Pd1, the majority of the genes exhibited a decreasing trend in gene expression over time during infection, with maximum values observed at 1 dpi. Notably, exceptions to this pattern were found in the cases of PDIP_11000, PDIP_40660, PDIP_66160, and PDIP_28970, where an increase in expression over time was observed, while PDIP_33600 and PDIP_22490 peaked at 2 dpi. In contrast, the expression for most of the Pd149 genes during fruit infection rose over time, albeit with values consistently lower than those observed for Pd1 (Figure 6).

The gene expression during in vitro growth exhibited notable distinctions between the two *P. digitatum* isolates. Generally, the gene expression levels in the low-virulent Pd149 isolate were higher than those observed during the infection process, although they followed a similar increasing trend during the time assayed. The highest expression level of the majority of the genes analyzed for the low-virulent isolate was detected after 3 days of incubation in PDB. In contrast, the levels of gene expression during the in vitro growth of Pd1 varied among genes. In some cases, the expression levels were higher than those detected during orange infection, while in others, the expression was clearly lower, highlighting the potential relevance of specific genes in the infection process (Figure 6).

Considering the induction levels reflected in Figure 5 and the expression pattern observed in Figure 6, 8 (PDIP_05570, PDIP_11000, PDIP_22490, PDIP_28970, PDIP_33600, PDIP_41920, PDIP_66160, and PDIP_74620) of the 21 selected genes did not display induction during infection with respect to the in vitro growth condition, at least in the initial stages (1 dpi). Of the remaining thirteen genes, two of them worth highlighting are PDIP_02280, which encodes a pectin esterase, and PDIP_76190, which encodes a hypothetical protein that seems to be a secreted thaumatin-like protein. The expression of these two genes was very high throughout the entire time and is almost exclusive during citrus infection by the Pd1 strain, peaking at 1 dpi.

Of note, other genes whose induction peaked at 1 dpi after infection in the Pd1 strain when comparing in vitro versus in vivo were PDIP_06990 (arginosuccinate lyase), PDIP_38040 (cell division control protein), PDIP_49600 (stress protein), PDIP_68700 (regulator of G protein), PDIP_76160 (Gti1/Pac2), and PDIP_78930 (protein Ca/calmodulin-dependent kinase). Induction extends from 1 to 2 days post infection in the case of PDIP_00580 (transcription factor C6), PDIP_01590 (LYR protein), PDIP_49640 (Snd1/p100), and PDIP_64910 (rhodanese protein). Only PDIP_40660, which encodes a facilitator of glycerol uptake, showed later induction during infection.

## 4. Discussion

Studies on fruit–pathogen interactions have greatly facilitated a better understanding of the pathogenicity and virulence of *P. digitatum* at the molecular level [2,28]. Nevertheless, as far as we know, no study has been conducted to identify *P. digitatum* genes putatively involved in virulence at early stages via the comparison of two strains differing in virulence degree. In this study, we have approached this topic by constructing a suppression subtractive hybridization (SSH) cDNA library enriched in *P. digitatum* genes that have higher expression in the more virulent strain.

Massive sequencing of the subtracted VPdS cDNA library showed that among the top 20 genes there are 5 that encode hypothetical proteins with unknown function and, curiously, there are 20 contigs that do not map to predicted genes in the annotated genome. GO annotation of the fungal genes indicated that the most represented biological processes are related to primary metabolism. KEGG pathways confirmed the importance of different enzymes and transporters, and signal transduction pathways represented by several transcription factors and regulatory proteins.

The transcriptional analysis carried out showed that selected genes were induced after 24 h in the virulent Pd1 isolate with respect to the low-virulent Pd149 isolate during the interaction with orange discs and during orange infection. This confirmed the validity of the subtractive approach and that the genes represented could probably play an important role in the differences observed between both isolates regarding their virulence.

Differential profile trends and magnitude were observed for all genes evaluated when comparing both strains in the presence of orange discs. In addition, in the virulent Pd1 strain, the expression of most of the selected genes was higher during the interaction with orange discs than during growth in vitro in PDB medium. However, given that the conditions used for the SSH only reflected the interaction but not the colonization of fruits, the study of the transcriptional profile was carried out for the selected genes both in vitro and during the infection of citrus fruits for both strains. Surprisingly, the gene with the greatest representation (PDIP_11000) showed the highest transcription rate in the virulent isolate during in vitro growth. This hypothetical protein contains a FluG domain, which has been described in the synthesis of a small diffusible factor that acts as an extracellular signal that directs asexual sporulation and perhaps other aspects of colony growth [38]. In fact, as shown in the characterization of both *P. digitatum* strains, apart from their great difference in virulence, a delay in sporulation could be observed in the case of Pd149. This could explain the huge difference in expression between both strains.

Although expression analysis showed that all genes had differential expression at 24 h in the virulent Pd1 isolate with respect to the low-virulent Pd149 strain (Figure 4 and Figure 5), only 13 (highlighted in bold in Table 2) of them were induced in Pd1 during citrus fruit infection as compared to in vitro growth. All of them also showed an expression profile with a decreasing trend over time. Of these 13 genes, the most induced genes corresponded to a pectin esterase (PDIP_02280) and a hypothetical protein that resembles a thaumatin-like protein (PDIP_76190) with an almost exclusive profile during infection. Interestingly, these two genes were also detected in another SSH cDNA library enriched in *P. digitatum* genes, which were highly expressed later during the infection process [18]. Moreover, PDIP_76190 was the gene with the highest induction magnitude in this latter study. The success of the infection and whether it progresses or not is determined at the first stages of the pathogen–host interaction and depends on the ability of the pathogen to penetrate and colonize the host tissues through the degradation of the polymers of the cell wall, among other routes. The role of cell wall degrading enzymes (CWDEs) in pathogenicity has been reported in postharvest pathogens such as *Alternaria alternata* [39], *Colletotrichum gloesporoides* [40], and *Penicillium expansum* [41]. The relevance of CWDEs was previously proved with the appearance of two pectin lyases (pnl1 and pnl2), a polygalacturonase (pg1), a pectin methyl esterase (PME), and three glucanases as predominant genes in the pathogenesis of *P. digitatum* [18] and in the appearance of different CWDEs genes during the initial infection process of *P. digitatum* in citrus [27,28]. Furthermore, it has been determined that fungal PMEs play a crucial role in the invasion of plant tissues during fungal infection by altering the pectin methyl esterification pattern [42]. An example of this is BCPME1, an important determinant of the virulence of *Botrytis cinerea* [43] and also a PME of *Fusarium graminearum*, which contributes to the virulence in wheat [42].

Ten genes showed their highest expression at 1 dpi, although in four of them, the induction extended to 2 dpi. The genes that stood out for their overexpression in the initial stages (24 h) were mostly associated with regulatory proteins. Notable among them were a regulator of the G protein signaling domain (PDIP_68700), a cAMP regulatory protein Gti1/Pact (PDIP_76160), and Ca/calmodulin-dependent protein kinase (PDIP_78930). Regulatory of G proteins signaling (RGS) has a conserved RGS domain that attenuates signal transduction, acting as negative regulators of G proteins, playing crucial roles in the growth, asexual development, and the pathogenicity of fungi. Thus, *rgsC* in *A. fumigatus* is involved in the control of vegetative growth, asexual sporulation, germination, stress response, gliotoxin production, and virulence [44]. In *Fusarium verticillioides*, two RGS, FlbA1 and FlbA2, differentially regulate the biosynthesis of fumonisin B [45], and in *Aspergillus flavus*, they regulate the formation of conidia, sclerotia, and aflatoxins production [46]. Gti1/Pac2 represents a conserved family of proteins that regulate toxin production and pathogenicity in the filamentous fungus *Fusarium graminearum* [47]. Both shared and distinct functions of Gti1/Pac2 protein families have been reported in the growth, morphogenesis, and pathogenicity of *Magnaporthe oryzae* [48]. In *Fusarium oxysporum*, the *FocSge1* gene, possessing a Gti-Pac2 domain, plays a critical role in normal growth and pathogenicity [49]. Ca^2+^/calmodulin-dependent protein kinases (CaMKs), a class of Ser/Thr protein kinases, mediate Ca^2+^ signals to modulate diverse biological processes. Fungal growth, sporulation, virulence, and stress tolerance are regulated through Ca^2+^-involved cross-linking mechanisms, including DNA replication and damage repair, cell cycle and nuclear division, and MAPK signaling. In *Penicillium italicum*, gene knockout and complementation analysis indicated the requirement of the PiCaMK1 in the fungal vegetative growth, sporulation, full virulence, and responses to salt (salinity) and mannitol (osmotic) stresses [50]. Additionally, in this group, although less represented, is argininosuccinate lyase (PDIP_06990). Argininosuccinate lyase (ASL) is part of the arginine biosynthetic pathway. Arginine is a relevant amino acid associated with cell signal transduction, protein synthesis, and sexual reproduction [51].

The other genes whose expression was maintained during days 1 and 2 post-inoculation included a putative rhodanese (PDIP_64910), a LYR protein (PDIP_01590), and two transcription factors (PDIP_00580 and PDIP_49640), respectively. Rhodanese is associated with cyanide detoxification as part of sulfur metabolism and can be considered important in anti-oxidative stress processes [52]. Proteins containing a LYR domain are an accessory subunit of the mitochondrial membrane respiratory chain NADH dehydrogenase [53]. Many transcription factors have been previously described for their role in regulating genes directly involved in fungal virulence. In *P. digitatum*, several genes encoding transcription factors have been identified, including different Zn_2_Cys_6_ transcription factors. Examples included *PdPacC*, which mediates responses to environmental pH and contributes to the pathogenicity of *P. digitatum* [17], *PdCrz1* encodes a calcineurin-sensitive transcription factor associated with the pathogenicity of *P. digitatum* [16], *PdSte12* affects conidiation and fungal virulence [22], and *PdMut3*, which is crucial for cell wall integrity and related to metabolism through peroxisomes degradation [25]. The transcription factor Snd1/p100 encoded by PDIP_49640 might directly regulate ergosterol metabolism by influencing the activation of sterol response element-binding protein 2 (SREBP2), potentially impacting lipid homeostasis in response to stress situations. Additionally, Snd1/p100 transcription factor, encoded by the *stf1* gene, plays a role in the sporulation regulatory mechanism of *Eurotium cristatum* [54].

Additionally, it is worth highlighting the genes that are overexpressed at 1 dpi but no later: a stress protein (PDIP_49600) and a cell division control protein (PDIP_38040). The chaperone-like protein CDC48 (cell division cycle 48) is an important component of ubiquitin-dependent protein degradation pathways and, more generally, of the protein quality control machinery. To survive in unpredictable conditions, all organisms must adapt to stressors by regulating adaptive cellular responses. Transcriptional control of the cell cycle by transcription factors is likely associated with the adaptation of the fungus to the host and the environment [55]. 

There was only one case (PDIP_40660), a glycerol uptake transporter whose maximum expression during infection occurred at a late infection stage (3 dpi). Fungal adaptation is necessary to endow the capacity to withstand stressful environments, such as the use of other carbon sources as energy. Glycerol plays a critical role in the function of infection of the rice blast fungus *M. oryzae* [56], and in some cases, sugar uptake transporters have been related to virulence, such as *PdSUT1*, which indirectly contributed to the virulence of *P. digitatum* [20].

In this sense, regulatory factors play a significant role in providing an advantage for better adaptation to the environment and stress. The developed fungal systems seem to involve mechanisms that improve access to nutrients either by wall degradation or by the transport of sugars and control of processes that affect development, sporulation, and infectivity.

## 5. Conclusions

This work highlights that the most virulent Pd1 isolate has increased the expression of enzymes such as CWDE (PME) and broad regulatory mechanisms that control fundamental aspects in terms of vegetative development, conidiation, and virulence. Different regulatory genes, ranging from transcription factors to protein kinases, underlie better adaptation, process control, and resistance to stressful environments.

## Figures and Tables

**Figure 1 jof-10-00235-f001:**
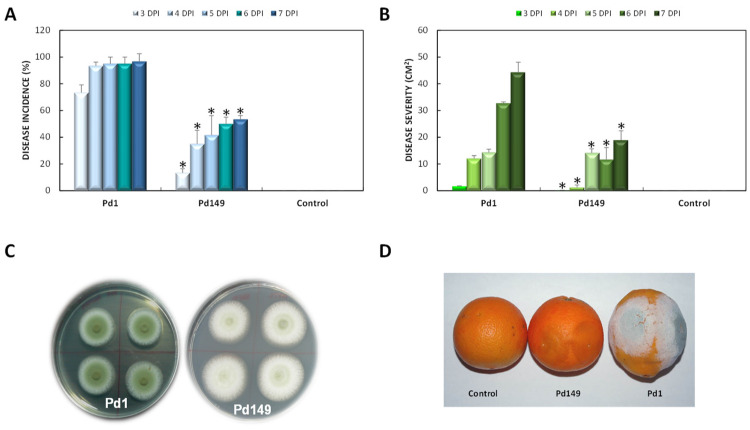
Comparison of fungal virulence between *P. digitatum* strains Pd1 and Pd149 on ‘Navelina’ oranges. (**A**): incidence of infection over time expressed in %. (**B**): severity of infection expressed in cm^2^. The results are the mean of two infection experiments. Error bars represent standard deviation. * Significant differences between treatments using Tukey’s test (*p* < 0.05) at each dpi. (**C**): Pd1 and Pd149 *P. digitatum* strains grown on PDA plates for 5 days. (**D**): Representative images of infected oranges at 6 dpi.

**Figure 2 jof-10-00235-f002:**
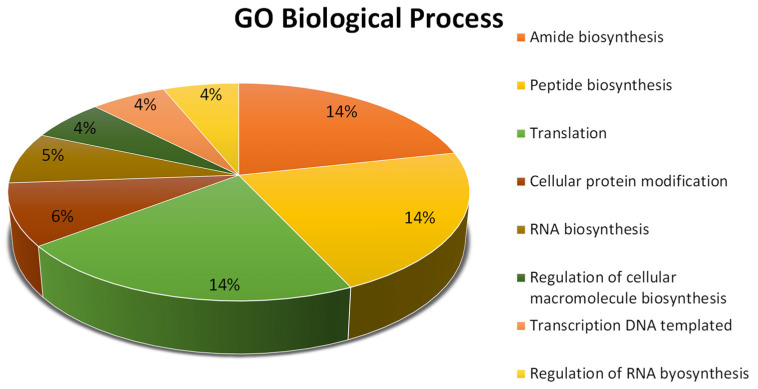
Gene ontology (GO) analysis of the total *Penicillium digitatum* contigs obtained in the subtracted VPdS cDNA library, representing the biological process. The chart shows the top 8 GO terms. Analysis was carried out using Blast2GO.

**Figure 3 jof-10-00235-f003:**
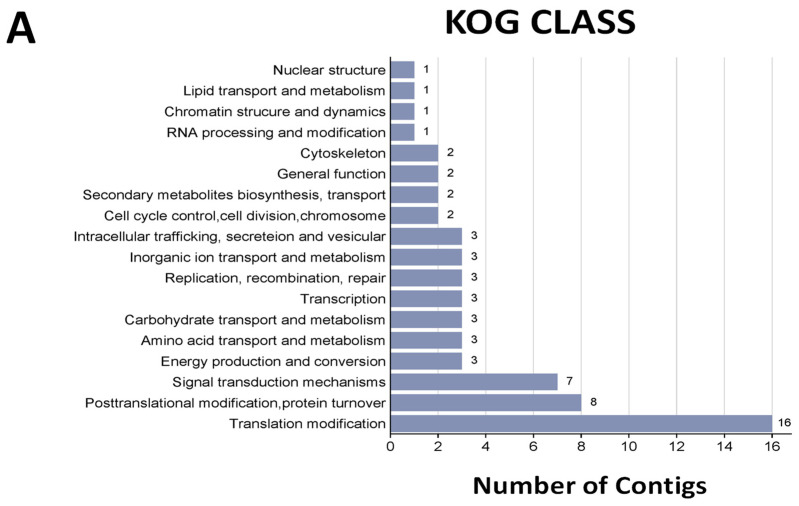
**Functional annotation is represented by different categories.** KOG class, KEGG pathways, and BRITE categories are sorted in increasing order according to the number of contigs.

**Figure 4 jof-10-00235-f004:**
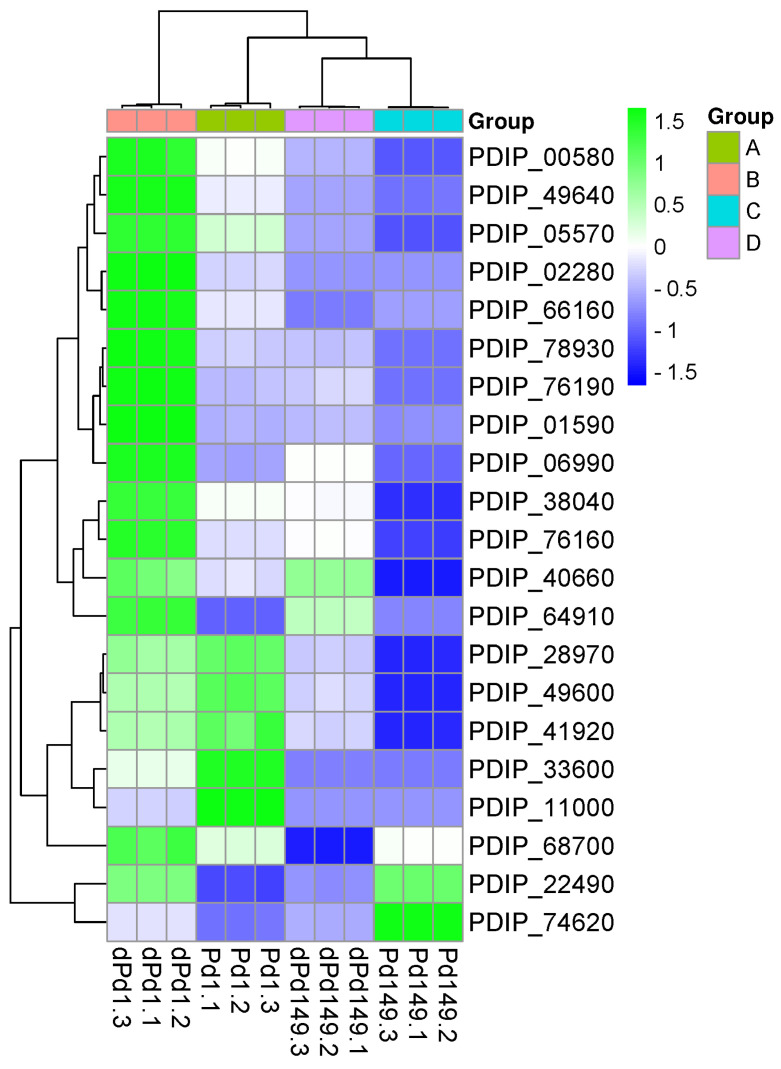
Heat map of the 21 selected genes across all treatments sorted by relative gene expression (RGE). Hierarchical clustering is included for the treatments (horizontal axis) and genes (vertical axis). Group A: Pd1(1-3) and group C: Pd149(1-3) correspond to three biological replicas of CECT 20795 and CECT 2954, respectively, grown for 24 h in PDB media on a 24-well plate. Group B: dPd1(1-3) and group D: dPd149(1-3) correspond to three biological replicas of CECT 20795 and CECT 2954, respectively, grown for 24 h interacting with orange discs (d). Color scaling was applied to the genes.

**Figure 5 jof-10-00235-f005:**
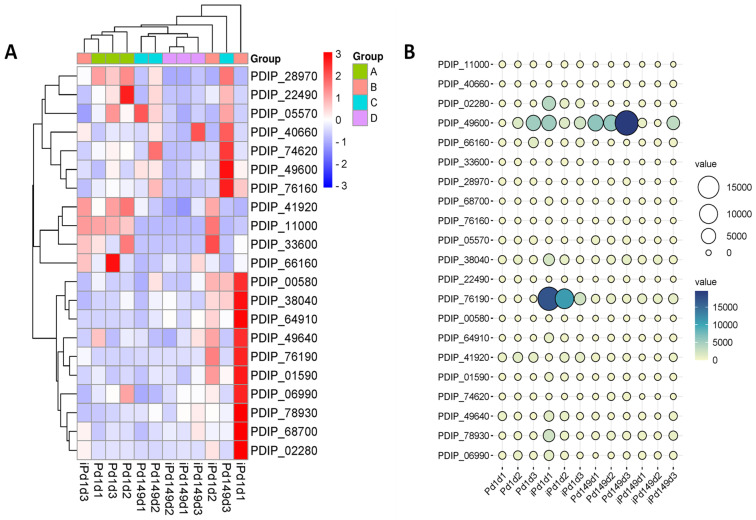
(**A**) Heat map showing 21 selected differentially expressed genes (DEGs) for in vitro, flasks with PDB medium, and in vivo (fruit infection) conditions sorted by RGE (Relative Gene Expression). Hierarchical clustering is included for the treatments (x axis) and for the genes (y axis). Color scaling was applied to the genes. (**B**) Ballon plot representing gene expression of 21 selected genes under different conditions over time. Group A and C correspond to Pd1 and Pd149, respectively, grown in PDB media in flasks. Group B and D correspond to Pd1 and Pd149, respectively, infecting orange fruits (i). In all cases, (d1–d3) correspond to 1, 2, and 3 days post inoculation (dpi). This figure concludes that virulent Pd1 isolate has raised the expression of enzymes such as CWDE (PME) and different regulatory mechanisms.

**Figure 6 jof-10-00235-f006:**
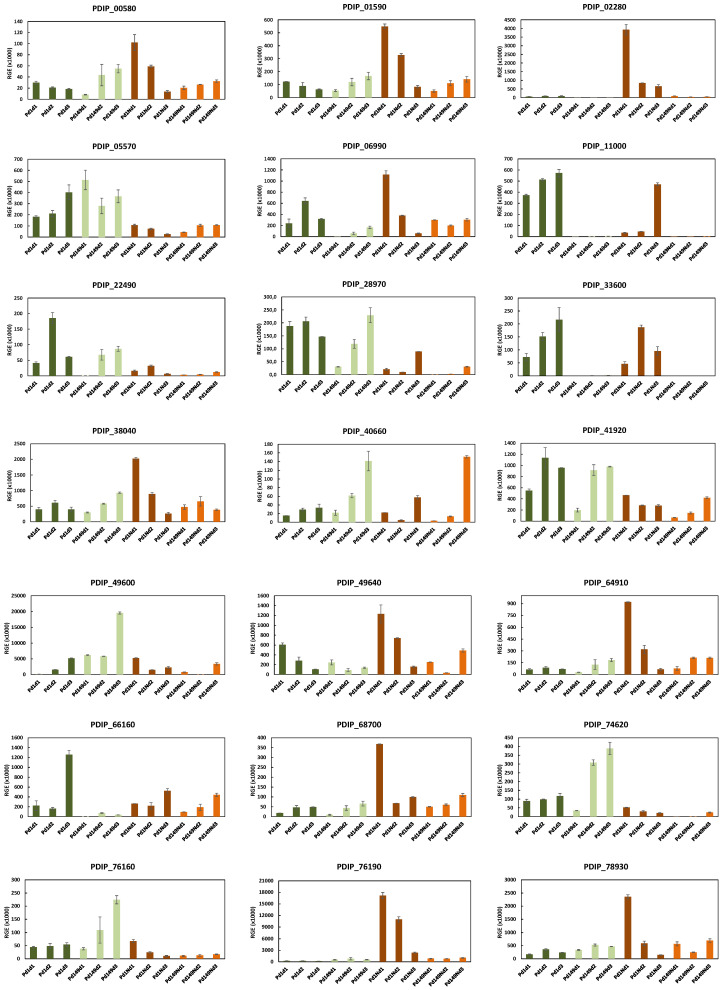
Analysis of relative gene expression (RGE) of 21 selected genes. Time course evaluation of gene expression in Pd1 and Pd149 strains grown in PDB liquid medium at 25 °C or during orange infection. In all cases, d1, d2, and d3 correspond to 1, 2, and 3 dpi, respectively. The expression levels are relative to β-tubulin as a reference gene. Error bars indicate standard deviations of three biological replicates.

**Table 1 jof-10-00235-t001:** Summary of the subtracted VPdS cDNA library.

Reads	81,354
Trimmed Reads	80,806
Aligned Reads	79,467
Assembled	58,387
Partial	6766
Singletons	792
Contigs	123
Isogroups	87
Isotigs	188

**Table 2 jof-10-00235-t002:** List of genes selected for further characterization and transcription analysis. Those genes whose expression was induced during infection are highlighted in bold.

ID	N° Reads	Description	Contig
PDIP_11000	12,951	Hypothetical protein-FluG protein domain	0020
**PDIP_40660**	**6882**	**Glycerol uptake facilitator**	0027
**PDIP_02280**	**4395**	**Pectinesterase family protein**	0049
**PDIP_49600**	**3737**	**Hypothetical protein-stress protein**	0084
PDIP_66160	3109	Hypothetical protein-Uncharacterized protein	0048
**PDIP_33600**	**2676**	**C6 transcription factor GAL4-like Zn (II)2Cys6**	**0074**
PDIP_28970	1252	Protein serine/threonine kinase (Ran1)	0003
**PDIP_68700**	**1233**	**Regulator of G protein signaling domain protein**	**0007**
**PDIP_76160**	**1222**	**cAMP-independent regulatory protein Gti1/Pac2**	**0034**
PDIP_05570	1096	Heat shock protein Hsp98/Hsp104/ClpA	0025
**PDIP_38040**	**1079**	**Cell division control protein cdc48**	**0010**
PDIP_22490	990	CorA family metal ion transporter	0062
**PDIP_76190**	**584**	**Hypothetical protein-Thaumatin like protein**	**0023**
**PDIP_00580**	**381**	**C6 transcription factor**	**0021**
**PDIP_64910**	**279**	**Hypothetical protein-Rhodanase protein**	**0013**
PDIP_41920	231	Iron copper transporter	0018
**PDIP_01590**	**150**	**LYR family protein**	**0026**
PDIP_74620	26	Mid2-like cell wall stress sensor	0068
**PDIP_49640**	**24**	**Transcription factor (Snd1/p100)**	**0050**
**PDIP_78930**	**15**	**Ca/calmodulin-dependent PK**	**0070**
**PDIP_06990**	**8**	**Argininosuccinate lyase**	**0078**

## Data Availability

Data are contained within the article and Appendix A.

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
