# Peer review of "Discovery and Transcriptional Profiling of Penicillium digitatum Genes That Could Promote Fungal Virulence during Citrus Fruit Infection"

_jof, 2024, doi:10.3390/jof10040235_

Round 1

Reviewer 1 Report

The research is adequately presented; however, the available evidence is insufficient.

The study conducted by Sánchez-Torres and colleagues unveiled multiple genes of Penicillium digitatum implicated in virulence during citrus infection through subtractive suppression hybridization. It is acknowledged that SSH is considered an outdated technology, prone to generating numerous false positive outcomes. Furthermore, the level of innovation in this research appears insufficient. While valuable genes may have been identified by the authors, there is a lack of direct evidence demonstrating their functions during the infection.  The results of gene disruption may be needed.

Author Response

We thank the reviewer for the comments. We consider that the approach chosen at the time in which the study was developed was the most promising and although today these approaches are made through RNAseq studies, it does not detract from the work carried out and the rigor in how it was carried out. As can be seen from the results, the selected genes all achieve the requirement of being differential in the selected situation. The innovation of the work lies in the comparison of two isolates with very different virulence at very short infection time such as 24 hours, while most early studies focus on 44 hours after infection. The use of two options such as discs and infected fruits highpoints the differences observed and highlights the need for the pathogen to have access to different nutrients.

Currently, many studies are based on a general study that allows for a global view of what is happening at a given moment, pointing to the possible mechanisms involved. In that sense, genetic disruption to demonstrate functionality is a useful but not always clarifying tool. On the other hand, some of the genes found have already been studied by other authors, which supports our hypotheses.

Reviewer 2 Report

The manuscript entitled “Discovery and transcriptional profiling of Penicillium digitatum genes that could promote fungal virulence during citrus fruit infection” by Sánchez-Torres et al. identified infection-related genes of Penicillium digitatum, an important citrus pathogen, by RNA-Seq. Overall, the experiment is well designed and the data is well organized. I recommend publication in JoF.

I have some concerns:

1. L230 and Table 2 “a total of 21 genes were selected : authors need to add more details on the reasons for selecting these 21 genes.

2. PDIP_11000 and PDIP_02280 and other most responsive genes during early infection, how about their gene function studies, authors could add some discussion in the these genes or their related family, are there any studies for the roles of them in virulence.

3. Penicillium digitatum is also a toxigenic fungi, well known for its production of mycotoxin (Patulin or OTA?), how about the expression profiles of these biosynthetic gene clusters during infection?

see above

Author Response

Many thanks to the reviewer for the comments and instructions. In response to the points made:

  1. L230 and Table 2 “a total of 21 genes were selected” : authors need to add more details on the reasons for selecting these 21 genes.

Many thanks to the reviewer for the comments and instructions. In response to the points made:

As mentioned in the text, the selection was founded on the one hand on the presence of a certain gene based on the number of reads, choosing the most represented ones. But on the other hand, genes were selected that, despite being poorly represented, had the hypothetical function assigned seemed interesting to study and delve into plausible mechanisms involved and how they could be related to each other.

In this sense, transcription factors, enzymes, transporters, etc. were selected that encompassed the genes with the greatest number of reads and the genes that had a medium to low number of reads but whose hypothetical function seemed to provide relevant information.

This point in the text has been modified to make it clearer.

  1. PDIP_11000 and PDIP_02280 and other most responsive genes during early infection, how about their gene function studies, authors could add some discussion in the these genes or their related family, are there any studies for the roles of them in virulence.

PDIP_11000 is described in the text as follows:

Surprisingly, the gene with the greatest representation (PDIP_11000) showed the highest transcription rate in the virulent isolate during in vitro growth. This hypothetical protein contains a FluG domain, which has been described in the synthesis of a small diffusible factor that acts as an extracellular signal that directs asexual sporulation and perhaps other aspects of colony growth [39]. In fact, as it shown in the characterization of both P. digitatum strains, apart from their great difference in virulence, a delay in sporulation could be observed in the case of Pd149. This could explain the huge difference in expression between both strains.

In the case of PDIP_02280, and other genes of interests previous studies are mentioned.

The most induced genes corresponded to a pectin esterase (PDIP_02280) and a hypothetical protein that resembles a thaumatin-like protein (PDIP_76190) with almost an exclusive profile during infection. Interestingly, these two genes were also detected in another SSH cDNA library enriched in P. digitatum genes highly expressed later during the infection process [18]. Moreover, PDIP_76190 was the gene with the highest induction magnitude in this latter study. The success of the infection and whether it progresses or not is determined at the first stages of the pathogen-host interaction and depends on the ability of the pathogen to penetrate and colonize the host tissues, through the degradation of the polymers of the cell wall, among others. The role of cell wall degrading enzymes (CWDEs) in pathogenicity has been reported in postharvest pathogens such as Alternaria alternata [40], Colletotrichum gloesporoides [41] and Penicillium expansum [42]. The relevance of CWDEs was previously proved with the appearance of two pectin lyases (pnl1 and pnl2), a polygalacturonase (pg1), a pectin methyl esterase (PME) and three glucanases as predominant genes in the pathogenesis of P. digitatum [18] and in the appearance of different CWDEs genes during the initial infection process of P. digitatum in citrus [27, 28]. Furthermore, it has been determined that fungal PMEs play a crucial role in the invasion of plant tissues during fungal infection by altering the pectin methyl esterification pattern [43]. An example of this is BCPME1, an important determinant of the virulence of Botrytis cinerea [44] and also a PME of Fusarium graminearum, which contributes to the virulence in wheat [43].

3.Penicillium digitatum is also a toxigenic fungi, well known for its production of mycotoxin (Patulin or OTA?), how about the expression profiles of these biosynthetic gene clusters during infection?

Penicillium digitatum is not considered a toxinogenic fungus and does not produce any toxin. Patulin is produced by Penicillium expansum. Ochratoxin A (OTA) is a mycotoxin produced by several species of fungi of the genera Penicillium and Aspergillus, mainly Penicillium verrucosum, Aspergillus ochraceus and especially Aspergillus carbonarius. For this reason, the clusters of these toxins were not evaluated in this study.

Reviewer 3 Report

This is a well-written paper. The focus of this study is to find the mechanisms underlineing the virulence at gene expression level. It is an excellent design to compare the difference for gene expressions between two isolates when growing in artificial media and the fruit tissue. I saw the differences between two isolates growing on plate and fruit. I am wondering if there is any difference for their spores to germinate in PDB and juice from the fruit since this study focused on the early stage of the infection process. Since CWDE was confirmed to be a major factor involved in the infection process, and though this is not something new, is there any difference between the two isolates for their production of pectin enzyme when growing in PDB and fruit tissue after 3 days, for example.

I am not an expert regarding the molecular work, but I like the data in Fig. 6 that I can see the difference easily. It will take a lot of analysis to understand some of the data, such as the heat map.

Author Response

Thanks for the reviewer's comments and suggestions.

The truth is that the differences in spore germination were not evaluated using different media to compare, which is a very interesting suggestion and we will take it into account for the future in the studies of both isolates. Nor was the production of pectinases evaluated between both isolates in the different media, only their differences at the expression level were evaluated, but it is something that we will take into account for future studies.

We understand that the simple representation is easier to evaluate than the heat map, but these also provide valuable information and that is why we have included them.

Round 2

Reviewer 1 Report

The research is adequately presented. While, some pictures should be improved.

Figure 1: The “STARS” in Figure 1A should be marked in the appropriate positions.

Figure 5 legend: a conclusion sentense should be added. 

Author Response

Thank you very much for your comments and following your instructions, Figure 1 has been modified and a conclusion sentence has been added to Figure 5 legend.
